# Association of the *TTN, PDK4,* and *RNF207* mutations with dilated cardiomyopathy in Dobermanns from the United Kingdom

**Luke C. Dutton**[1]*, **Andrew Crosland**[2], **Joanna Dukes-McEwan**[2], **David J. Connolly**[1]

1 Department of Clinical Science and Services, Royal Veterinary College, Hatfield, London, United Kingdom, 2 Department of Small Animal Clinical Science, University of Liverpool, Leahurst Campus, Neston, United Kingdom

* lcdutton@rvc.ac.uk

## Abstract

A missense mutation in the titin gene (*TTN*) and a splice-site mutation in the pyruvate dehydrogenase kinase 4 gene (*PDK4*) have been associated with dilated cardiomyopathy (DCM) in Dobermanns from the United States. Additionally, a missense mutation in the gene *RNF207* has been reported in association with DCM from a European Dobermann cohort. Based on this we examined the association of these variants with DCM in United Kingdom (UK) Dobermanns. We hypothesized that the *TTN* and *PDK4* gene variants would not be associated with DCM in UK Dobermanns and that there would be an association between the *RNF207* mutation and DCM. We included 74 client owned dogs (30 control dogs and 44 dogs with DCM) in the study. Allele frequencies for each variant were calculated. Chi-square testing was used to assess for differences in allele frequencies and genotype proportions between groups. Overall allele frequency in this cohort was 35% for the *TTN* variant, 18% for the *PDK4* variant, and 37% for the *RNF207* variant. There was no difference in allele or genotype frequencies between control and DCM dogs for *TTN* or *PDK4* (p = 0.79 for both allele frequencies, p = 0.91 for *TTN* and p = 0.78 for *PDK4* genotype frequencies). There was a significant difference in the allele frequencies of the *RNF207* variant between DCM cases and controls (OR 2.4 (95% CI 1.07 − 5.15), p = 0.03) and genotype frequencies for *RNF207*, with a homozygous genotype found almost exclusively in DCM dogs (p = 0.034). We conclude that the previously reported *RNF207* variant appears associated with DCM in UK Dobermanns, but there was no association with the previously reported *TTN* or *PDK4* mutations. This is important when considering selective breeding in different populations of Dobermanns. However, the small sample size may impact the generalizability of the results.

## Introduction

Dilated cardiomyopathy (DCM) is the most common cardiomyopathy in dogs with a strong breed predisposition, resulting in significant morbidity and mortality [1,2]. In Dobermanns this autosomal dominant disease had a cumulative prevalence of almost 60% in European dogs, 40% in North American dogs, and can result in congestive heart failure or sudden cardiac death [3,4].

**Data availability statement:** All data files are available from the figshare database (https://doi.org/10.6084/m9.figshare.28248788.v1)

**Funding:** The author(s) received no specific funding for this work.;

**Competing interests:** The authors have declared that no competing interests exist.

In humans, > 100 genes have been implicated in inherited DCM [5,6]. Studies in Dobermanns from the United States (US) have indicated a single nucleotide polymorphism (SNP) in the titin gene (*TTN*) and splice-site mutation in the pyruvate dehydrogenase kinase 4 gene (*PDK4*) as being associated with DCM [7–9]. However, studies in European-bred Dobermanns have failed to confirm this association [10,11]. US-bred and European-bred Dobermanns represent separate genetic pools, therefore it is possible that causative mutations identified in one geographical location are not disease associated in another [12]. However, it may also reflect the polygenetic features of DCM and challenges in assigning genotype-phenotype associations, which raises issues when designing genetic screening tests for breeding programs.

In a recent study of a large cohort of European Dobermanns (predominately from Germany), no association between the *TTN* and *PDK4* mutations and DCM was identified. However a possible loci on chromosome 5 was suggested to be involved in the development of a DCM phenotype [11]. Building on work from a previous study [13], the team went on to find a SNP in the *RNF207* gene and a deletion in the mRNA sequence for *PRKAA2* that appeared associated with DCM. However, no DNA sequence alteration was identified in *PRKAA2* that would explain the mRNA deleted sequence. *RNF207* encodes a heart-specific really interesting new gene (RING) finger protein that has been shown to interact with the voltage-dependent anion channel (VDAC) 1, regulate the repolarizing channel HERG and is involved with pathological cardiac hypertrophy [14–16]. Therefore, variants in *RNF207* may result in QT prolongation, shortened action potential duration and altered energy metabolism, which may play a role in DCM.

The previously reported *TTN* and *PDK4* have failed to show an association with DCM in European Dobermanns and the newly discovered *RNF207* mutation has not been studied in a UK cohort of Dobermanns. Our hypothesis was that the *TTN* and *PDK4* gene variants would not be associated with DCM in a UK cohort of Dobermanns and that there would be an association between the more recently identified *RNF207* SNP and DCM in UK Dobermanns. We aimed to sequence the suspected causative loci within these genes in a cohort of Dobermanns both with and without evidence of DCM.

## Materials and methods

### Animals

Multicenter, retrospective case-control study. The study was approved by the Royal Veterinary College ethical review committee (URN: M2020 0163). The major cohort was obtained from the Centre for Integrated Genomic Medical Research (CIGMR), University of Manchester, that archived DNA samples from Dobermanns that participated in PROTECT and LUPA projects [17,18] and also Dobermanns presenting for screening or as clinical cases to the University of Liverpool Small Animal Teaching Hospital, whose owners had consented to surplus blood, after clinically indicated blood tests, being stored in the DNA archive. A further cohort included dogs owned by members of the Dobermann breed club in the UK that had been screened by an accredited veterinary cardiologist for DCM. These owners were requested to obtain a saliva sample from their dog using the PERFORMAgene PG-100 collection kit according to the manufacturer's instructions (DNA Genotek Inc., Canada), and submit this along with the DCM screening test results. Additionally, samples were obtained from Dobermanns presenting to the Queen Mother Hospital for Animals, Royal Veterinary college, that had investigations for DCM during their hospital visit, with owner consent to store residual blood samples.

Dilated cardiomyopathy was diagnosed in the majority of dogs according to the European Society of Veterinary Cardiology (ESVC) screening guidelines for Dobermanns [19], with

some dogs being included based on the inclusion criteria for the PROTECT study (prior to screening guidelines) [15]. Dogs were diagnosed with DCM if they had > 300 ventricular premature complexes on a 24-hour Holter monitor and/or they had echocardiographic evidence of DCM. Echocardiographic criteria were an end-diastolic volume indexed to body surface area of > 95 mL/m$^2$ or end-systolic volume indexed to body surface area of > 55 mL/m$^2$. Alternatively, if only M-mode measurements of the left ventricle were obtained, a left ventricular end-diastolic diameter of > 48mm (male), > 46mm (female) or left ventricular end-systolic diameter of > 36mm (male or female) were used to diagnose DCM. Dogs with DCM could be any age at the time of diagnosis and present with occult DCM (either echocardiography positive or Holter positive or both) or present with evidence of congestive heart failure. Dobermanns were included as controls if they were > 7 years old without echocardiographic evidence of DCM (according to the ESVC screening guidelines listed above or a specialist cardiologist reporting a structurally normal heart) at time of sample collection. If Holters were performed then they had to have < 50 ventricular premature complexes (VPCs) in 24 hours and, if no Holter was performed, then follow-up was undertaken by telephone to confirm a cause of death other than cardiac.

**DNA extraction.** For samples requiring DNA extraction the following protocols were followed. The DNA from collected saliva samples was purified according to the manufacturer's instructions up to the formation of a DNA pellet. DNA pellets from saliva samples and blood samples were then processed using an on-column purification kit (DNeasy Blood and Tissue Kit, Qiagen Ltd) according to the manufacturer's instructions. DNA was quantified using a spectrophotometer (DS-11 Series, DeNovix Inc.) and a 260/280 ratio of ≈ 1.8 taken to indicate good quality DNA.

**Primer design.** Primers to amplify the *PDK4* loci were already published [9]. Primers for the *TTN* mutation and *RNF207* mutation were designed using Primer–BLAST (National Library of Medicine, MD, US) and are shown in Table 1. All primers were from Merck (Merck Life Science UK Limited, Dorset, UK) and reconstituted to a concentration of 200 pmol µL$^{-1}$ using molecular biology grade water. Forward and reverse primers were mixed and diluted to 10 pmol µL$^{-1}$ then stored at - 20°C.

**Polymerase chain reaction and sequencing.** Reactions contained 1 µL of template genomic DNA or water controls. In addition, 2 µL of PCR primers, MyTaq™ DNA polymerase (Meridian Bioscience, London, UK), MyTaq™ reaction buffer (Meridian Bioscience), and an appropriate amount of molecular biology grade water was added to make up reaction volumes to 25 µL in a final concentration recommended by the manufacturer (Meridian Bioscience). PCR was performed using a thermal cycler (Applied Biosystems™ SimpliAmp™ Thermal cycler, Thermo Fisher) set to run the following cycling settings: 95 °C for 3 min, followed by 35 cycles of 95 °C for 15 s, 55 °C for 20 s and 72 °C for 30 s, with a final extension at 72 °C for 10 min. Successful amplification was assessed using a 2% gel electrophoresis. PCR products for sequencing underwent post-PCR clean-up using the GenElute™ PCR Clean-Up Kit (Merck

**Table 1. List of primers used in this study.**

| Gene | Sequence (5'-3') | Product size (bp) |
|---|---|---|
| *TTN* | F: TTCCAACCCTATGGGTACGTT | 307 |
| | R: GCTGACGAGTGACAGCAAAT | |
| *PDK4* | F: TTCTTTGCCAGTAACTGATCTTT | 249 |
| | R: TGCATGGACTCTCTCTCTCTCA | |
| *RNF207* | F: GAGGACACAGCCTTCACA | 722 |
| | R: TCGTAAATCTCCTGTTCGTT | |

Life Science UK Limited, Dorset, UK) according to the manufacturer's instructions. Samples were then sent for standard Sanger sequencing (Eurofins Genomics, Ebersberg, Germany) using the forward primer as a template. Sequences were then aligned to the consensus sequence (CanFam3.1) using Genious Prime for MacOS (Biomatters Inc. Boston, US). Each sample was then assessed as to whether it was wild-type (-/-), heterozygous (+/-) or homozygous (+/+) for either the *PDK4*, *TTN* or *RNF207* variants.

**Statistical analysis.** Statistical analysis was performed using commercially available software (GraphPad Prism version 9.3.1 for Mac OS X, GraphPad Software, San Diego, California USA). Data was assessed for normality using the Shapiro-Wilk test. Parametric data are presented as mean (standard deviation) and non-parametric data presented as median (range). Categorical data is presented as frequency and percentage. Comparison between two groups were carried out using student's T-test for continuous parametric data or Mann-Whitney U test for non-parametric data. Chi-square test was used to compare allele frequencies and genotype frequencies between groups. P values < 0.05 were considered statistically significant.

## Results

### Study population

The CIGMR database consisted of 376 Dobermanns, of which 54 Dobermann samples (35 with DCM and 19 controls) were selected and the others excluded for reasons including unaffected dogs being < 7 years old, or without a Holter examination or echocardiographic data present, or no final diagnosis made/recorded in the database. The rest of the study group comprised nine Dobermanns presenting to the QMHA where residual blood was stored and 12 saliva samples from dogs owned by members of the Dobermann breed club in the UK. Therefore, in total our study population consisted of 74 Dobermanns, comprising 30 control dogs and 44 Dobermanns with DCM. The median age was 8.0 years (1.0 – 12.0 years). There was no difference in the age between control dogs and DCM dogs (median 8.1 years (7.0 – 12.0) versus 7.9 years (1.0 – 11.0), p = 0.094). Female dogs were overrepresented in the total population (58%) and there was no difference in sex distribution between the groups (p = 0.77). The mean bodyweight was 38.0 kg ( ± 5.2). There was no difference in body weight between the two groups (p = 0.69). Out of the dogs with DCM, 19/44 (43%) presented in congestive heart failure and 5/44 (11%) experienced a documented sudden death.

**Echocardiographic and electrocardiographic data.** The end-diastolic volume index (EDVI) in the DCM group was greater than the control group (119.7 mL/m$^2$ ( ± 42.4) versus 64 mL/m$^2$ ( ± 13.6), p < 0.001) as was the end-systolic volume index (84.2 mL/m$^2$ ( ± 43) versus 29.2 mL/m$^2$ ( ± 7.2), p < 0.001). Similarly, the left ventricular internal dimension in diastole (LVIDd) was larger for the DCM group compared to the control group (56.3 mm ( ± 11.2) versus 40.6 mm ( ± 3.0), p < 0.001) and left ventricular internal dimension in systole (LVIDs) was larger for the DCM group compared to the control group (44.3 mm ( ± 18) versus 29.0 mm ( ± 3.5), p = 0.011). Of the dogs with DCM, 33 dogs were diagnosed based on echocardiographic measurements alone, two dogs diagnosed based on Holter analysis alone and 9 dogs diagnosed based on a combination of echocardiography and Holter analysis. Where data was recorded on arrhythmia burden (n = 13), the median number of VPCs detected in dogs with DCM was 1327 VPCs/24 hours (3 – 18,969).

**Allele frequencies for *TTN*, *PDK4* and *RNF207* variants.** Allele frequencies are shown in Table 2. In our population of Dobermanns, 31 were wild-type (WT), 31 heterozygous and 12 were homozygous for the *TTN* variant, giving an overall allele frequency of 37%. The *PDK4* variant was present in 8 heterozygous, 9 homozygous, and 57 dogs were wild-type at this loci,

**Table 2. Allele frequencies for each variant and association with DCM.**

| | | Control dogs | | DCM dogs | | | |
|---|---|---|---|---|---|---|---|
| Gene variant | *n* | – | + | – | + | p-value | OR |
| *TTN* | 74 | 0.51 | 0.49 | 0.54 | 0.46 | 0.79 | 0.9 |
| *PDK4* | 74 | 0.27 | 0.73 | 0.30 | 0.70 | 0.79 | 1.1 |
| *RNF207* | 74 | 0.64 | 0.36 | 0.43 | 0.57 | **0.03** | **2.4** |

giving an allele frequency of 18%. For the *RNF207* variant, 33 dogs were WT, 29 dogs were heterozygous, and 12 dogs were homozygous, giving an allele frequency of 37%.

**Association of the TTN and PDK4 variants with dilated cardiomyopathy.** Of the dogs with DCM, 19/44 (43%) were wild type (WT), 18/44 (41%) were heterozygous, and 7/44 (16%) were homozygous for the *TTN* variant. In the control dogs, 12/30 (40%) were wild type (WT), 13/30 (43%) were heterozygous, and 5/30 (17%) were homozygous for the *TTN* variant. There was no significant difference in the allele frequencies of the *TTN* variant between cases and controls (Table 2 and Fig 1A, OR 0.9 (95% CI 0.45 – 1.86), p = 0.79). When the *PDK4* variant was sequenced, of the DCM dogs 34/44 (77%) were wild-type (WT), 4/44 (9%) were heterozygous, and 6/44 (14%) were homozygous, and for the control dogs 23/30 (77%) were wild-type (WT), 4/30 (13%) were heterozygous, and 3/30 (10%) were homozygous. There was no significant difference in the allele frequencies of the *PDK4* variant between cases and controls (Table 2 and Fig 1B, OR 1.14 (95% CI 0.47 – 2.90), p = 0.79).

**Association of the RNF207 variant with dilated cardiomyopathy.** Of the dogs with DCM, 13/44 (30%) were wild type (WT), 16/44 (36%) were heterozygous, and 11/44 (25%) were homozygous for the *RNF207* variant. In the control dogs, 14/30 (47%) were wild type (WT), 13/30 (43%) were heterozygous, and 1/30 (3%) were homozygous for the *RNF207* variant. There was a significant difference in the allele frequencies of the *RNF207* variant between cases and controls (Table 2 and Fig 1C, OR 2.4 (95% CI 1.07 – 5.15), p = 0.03). There was no difference in the age of diagnosis of DCM between dogs that were homozygous for

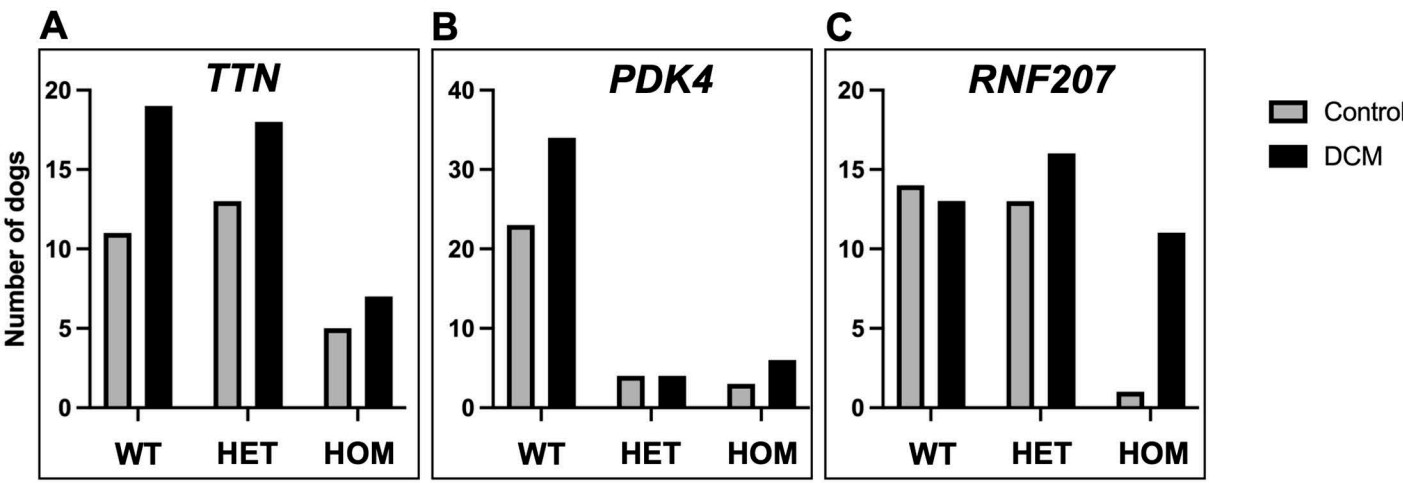

**Fig 1. Genotypes for control dogs and dogs with dilated cardiomyopathy (DCM) for each of the gene variants investigated.** There was no difference in genotype frequencies between wild-type (WT), heterozygous (HET), and homozygous (HOM) dogs between the control and DCM group for the *TTN* variant (A, **p** = 0.91) or the *PDK4* variant (B, **p** = 0.78). There was a difference in the frequencies of genotypes for the *RNF207* with more homozygous dogs in the DCM group (C, **p** = 0.03). Grey bars = control dogs and black bars = DCM affected dogs. WT = wild-type, HET = heterozygous, and HOM = homozygous for the variant at that specific loci.

*RNF207* and those that were heterozygous or WT (p = .44). The one control dog that was homozygous for *RNF207* was screened at 7 years old but lived to 11 years and 9 months, although further screening results and cause of death were not known.

## Discussion

In this study we aimed to assess the association of the previously reported *TTN*, *PDK4*, and *RNF207* variants with DCM in a population of Dobermanns from the UK. In our study population, whilst the *TTN* and *PDK4* variants were present, there was no association with the development of DCM. However, the *RNF207* variant did appear associated with DCM, and a homozygous genotype was almost exclusively seen in UK Dobermanns diagnosed with DCM.

In humans, there are >100 genes that have been associated with the development of inherited DCM [6,20]. Depending on the study, genetic testing can identify a known associated variant in around 50% of people, which has led to improved genetic counselling, screening, and treatment for affected families and individuals [6,20,21]. For example, mutations in *LMNA* and *PLN* have been associated with a higher prevalence of ventricular arrhythmias and sudden cardiac death than those in sarcomeric genes [5]. In contrast, only relatively few genetic mutations have been identified and associated with familial DCM in dogs. In Boxer dogs, DCM has been linked to a deletion in the striatin gene [22], in Irish Wolfhounds six loci have been identified using a genome-wide association study [23], and in Dobermanns there have been loci identified on chromosome 5, 14 and 36 [7–9,13]. Studies have also ruled out an association with selected other known human mutations [24–27].

Selective breeding over hundreds of years narrow the genetic pool in purebred dogs and amplify disease causing or modifying mutations. This makes dogs an excellent large animal model to study human disease, especially in DCM where there are striking phenotypic similarities between humans and dogs [28–30]. Dobermanns have been of particular interest for DCM related genetic studies due to the high prevalence, well characterized disease, and narrow genetic pool [11,12]. It is important to note that the worldwide Dobermann population exist as four major cohorts, two European and two American, therefore it may be possible that different loci are associated with disease in different cohorts [12]. In Dobermanns from the US, one mutation located on chromosome 14 and another on chromosome 36 have been identified as being associated with DCM [7,9]. A 16-bp deletion in the gene encoding pyruvate dehydrogenase kinase 4 (PDK4), reported by Meurs *et al.* in 2012, was found to be associated with DCM in Dobermanns [9]. PDK4 has a critical role in energy metabolism, therefore it was thought that dogs with this mutation would have a reduced ability to regulate glucose oxidation, causing an energy starved state [9,31,32]. However, there was no association of this mutation with DCM in a larger European cohort of Dobermanns, and it was also identified in other dog breeds, questioning whether this mutation is truly associated with DCM [10]. Our study had similar findings, in that we did not identify an association between the *PDK4* mutation and DCM in our cohort, and we report a similar allele frequency to the European population. It is possible that regional breeding practices in the UK may have influenced the prevalence of DCM-associated mutations compared to other populations, and given the limited genetic diversity of Dobermanns in the UK there is the possibility of founder effects. However our study did not have a sufficient sample size to assess if geographical location affected a genotype-phenotype relationship.

A missense variant in the gene encoding titin (*TTN*) has also been reported in a US cohort of Dobermanns [7,8]. This C to T base change results in a highly conserved glycine residue changing to arginine, which was predicted to be damaging by computer models [7]. Titin is the largest protein in humans, made up of around 30,000 amino acids. In the heart it is bound to both the thick and thin filaments, playing a major role in the structure of the sarcomere,

signal transduction, mechano-sensing, and contraction/relaxation kinetics of the myocardium [33,34]. Specifically, titin acts as a so-called molecular spring, where it contributes to the stiffness of the myocardium as well as contraction [35]. In humans, variants in *TTN* that result in a truncated protein account for 20–25% of DCM cases where a genetic mutation is found [33,36]. They are reported to occur throughout the titin gene, with most being found in the region of titin that spans the A-band of the sarcomere [37]. The variant in *TTN* that was discovered by Meurs *et al.* was instead located in an immunoglobulin-like domain of titin called I71 and is part of one of the spring elements that make up the protein [38]. It is hypothesized that the missense mutation in this I71 may unfold the region or make it easier to unfold during diastole, which may make it sensitive to degradation and affect passive and/or active tension [7]. However, although reported in a US population of Dobermanns, we did not find an association between the *TTN* SNP and DCM in our cohort of dogs, and a GWAS performed in a large European Dobermann cohort also did not find association between loci on chromosome 14 (*PDK4*) or chromosome 36 (*TTN*) [11].

In 2011, Mausberg *et al.* identified a loci on chromosome 5 that appeared to be linked with DCM in Dobermanns, but at that time a specific mutation was not identified [13]. This was followed up by a larger GWAS study of > 500 Dobermanns with a European background in 2023 that confirmed a disease-associated loci on chromosome 5 [11]. The authors further identified a specific SNP in the *RNF207* gene located on chromosome 5 that was associated with DCM. They also found association with another gene, *PRKAA2*, however no clear candidate genomic variant explained the aberrant splicing seen with RNA transcripts of this gene. In our study we also found an association between the *RNF207* variant and DCM, with the majority of dogs that were homozygous for the variant presenting with a DCM phenotype. *RNF207* encodes the really interesting new gene (RING)-finger protein 207 (RNF207), which has proposed actions in regulating the cardiac action potential and is involved in pathways known to regulate cardiac hypertrophy [15,16]. Additionally, it has been shown in humans that abnormal function of the RNF207 protein is implicated in or can worsen the expression of long-QT syndrome, known to result in ventricular arrhythmias causing syncope and sudden death [39,40]. Also it has been shown that RNF207 plays an important role in the production of ATP in cardiomyocytes, and that dysfunction in this protein may worsen the development of congestive heart failure [14]. In a study using canine myocardium, researchers discovered that RNF207 is expressed in the intercalated discs between cardiomyocytes (as well as in the cytoplasm and perinuclearly) [11]. Since the gap junctions present in the intercalated discs are vital for ion passage between cells and coordinated cardiac contraction, disruption to this region may lead to impaired cardiac function. Other studies have shown that alteration to the intercalated discs is directly associated with the development of DCM [41–43]. Additionally, immunohistochemistry in a dog that was homozygous for the *RNF207* variant showed marked cellular mosaicism for RNF207, indicating the variant alters expression of the protein that may affect cardiac function by the previously mentioned mechanisms [11]. Additionally, due to the role of *RNF207* in ATP production in cardiomyocytes, it is conceivable that mutations in *RNF207* could influence the expression of DCM in Dobermanns as opposed to being directly causal, and this would also explain why not all Dobermanns with DCM carry this variant. It is therefore also possible that there are specific subgroups or characteristics of the dogs with the *RNF207* variant that may influence the strength of the association with DCM, or specific phenotypic features, that were unable to be identified in this study and would benefit from future investigation.

Our study has a number of limitations. First, selecting a control population in a breed with a very high prevalence of a disease that displays age-related penetrance is challenging. We selected 7 years old as an age cut-off for our control population of dogs as it has been shown

that if dogs are normal at this examination, they are unlikely to go on to develop DCM [3]. Additionally, 7 control dogs did not have Holter examinations at the time of screening, therefore although other causes of death were reported, these dogs may still have had occult (i.e., non-clinical) DCM, as around 15–20% of dogs in this age group may have only VPCs detected on screening [3]. Therefore, it is possible our control population contained a small number of dogs with DCM. Additionally, as this was a retrospective study, not all echocardiographic and/or Holter data was available for review, but cases could still be included if they had an examination with a specialist cardiologist and they had been deemed affected or unaffected. The cases and controls from the CIGMR DNA archive preceded the recent guidelines for Dobermann DCM [19]. Echo and Holter data were reviewed where available, to ensure these criteria were met. However, it is possible that the diagnosis of normal or DCM was made by cardiologists based on historical guidelines, including the PROTECT study inclusion criteria [18]; these are considered by the authors to be robust. The sample size was small for a genetics study, which may limit the generalizability of the results, and lead to type II errors where true associations are missed, which are important factors when interpreting our findings. Studies that are prospective using a larger population of Dobermanns would therefore be needed to confirm these findings. Additionally, the small sample size and possibility that sampling did not represent the UK population as a whole, there is potential for bias in the sampling method (which relied on owners presenting dogs for examination rather than random sampling) and representativeness of the sample to the UK as a whole. Related to this, assessment of geographical location within the UK could help decipher whether there are founder effects resulting in differences between geographically distinct Dobermann populations.

Overall our study found no association with the previously reported US Dobermann DCM-associated variants in *TTN* or *PDK4*, but did find an association with the *RNF207* variant. This is in agreement with a study on European Dobermanns predominately from Germany [10,11,13]. It is currently unclear if this represents population-based differences in DCM-associated genes between US and European Dobermanns. However, this seems less likely given the remarkable phenotypic similarities in DCM between these populations and that a large proportion of US Dobermanns are direct descendants of 7 dogs imported from Germany in 1941 [44]. Regardless it highlights the difficulties in exploring a disease that has complex genetics and variable penetrance in a population. Caution is advised when using variants in genetic testing to screen Dobermanns and base selective breeding on these results, as this may only result in a more restricted genetic pool. Further research is warranted that validates the findings for the variant in *RNF207* using larger sample sizes with prospective study design and following dogs longitudinally to assess outcome in patients with certain genotypes. Additional research could then assess whether early detection of at-risk individuals and therefore earlier clinical intervention (such as more regular screening or treatment with disease-modifying drugs) may improve outcomes in this patient population.

## Acknowledgements

The authors would like to thank Sue Thorn and the members of UK Dobermann breed clubs for participating in this research study and providing saliva samples for DNA extraction and analysis.

## Author contributions

**Conceptualization:** Luke C. Dutton, Joanna Dukes-McEwan, David J. Connolly.

**Data curation:** Luke C. Dutton, Andrew Crosland, Joanna Dukes-McEwan, David J. Connolly.

**Formal analysis:** Luke C. Dutton, Joanna Dukes-McEwan.

**Investigation:** Luke C. Dutton, Joanna Dukes-McEwan.

**Methodology:** Luke C. Dutton, Joanna Dukes-McEwan, David J. Connolly.

**Project administration:** Luke C. Dutton.

**Supervision:** Joanna Dukes-McEwan, David J. Connolly.

**Writing – original draft:** Luke C. Dutton.

**Writing – review & editing:** Luke C. Dutton, Andrew Crosland, Joanna Dukes-McEwan, David J. Connolly.

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
