## [Decision Letter · Decision Letter 0]

6 Nov 2024

PONE-D-24-45194Association of the TTN, PDK4, and RNF207 mutations with dilated cardiomyopathy in Dobermanns from the United KingdomPLOS ONE

Dear Dr. Dutton,

Thank you for submitting your manuscript to PLOS ONE. After careful consideration, we feel that it has merit but does not fully meet PLOS ONE’s publication criteria as it currently stands. Therefore, we invite you to submit a revised version of the manuscript that addresses the points raised during the review process.

We look forward to receiving your revised manuscript.

Kind regards,

Nejat Mahdieh

Academic Editor

PLOS ONE

Journal Requirements:

Please confirm at this time whether or not your submission contains all raw data required to replicate the results of your study. Authors must share the “minimal data set” for their submission. PLOS defines the minimal data set to consist of the data required to replicate all study findings reported in the article, as well as related metadata and methods (https://journals.plos.org/plosone/s/data-availability#loc-minimal-data-set-definition ).

If your submission does not contain these data, please either upload them as Supporting Information files or deposit them to a stable, public repository and provide us with the relevant URLs, DOIs, or accession numbers. For a list of recommended repositories, please see https://journals.plos.org/plosone/s/recommended-repositories .

3. Please upload a copy of Figure 2B to which you refer in your text on page 11. If the figure is no longer to be included as part of the submission please remove all reference to it within the text.

Reviewers' comments:

Reviewer's Responses to Questions

**Comments to the Author**

1. Is the manuscript technically sound, and do the data support the conclusions?

Reviewer #1: Yes

Reviewer #2: Yes

2. Has the statistical analysis been performed appropriately and rigorously? 

Reviewer #1: Yes

Reviewer #2: Yes

3. Have the authors made all data underlying the findings in their manuscript fully available?

Reviewer #1: Yes

Reviewer #2: Yes

4. Is the manuscript presented in an intelligible fashion and written in standard English?

Reviewer #1: Yes

Reviewer #2: Yes

5. Review Comments to the Author

Reviewer #1: Abstract

The abstract presents a study investigating the association of specific genetic mutations with dilated cardiomyopathy (DCM) in Dobermanns, focusing on three genes: TTN, PDK4, and RNF207. The authors hypothesized that the TTN and PDK4 mutations would not be associated with DCM in a UK cohort, while expecting an association for the RNF207 mutation. The study involved 74 dogs, including both controls and DCM-affected animals.

Undoubtedly, the hypothesis is clearly stated, providing a solid foundation for the study and the methods are appropriate for the research question. In addition, the results are presented in a concise manner, highlighting the key findings regarding allele frequencies and statistical significance and finally in conclusion you emphasize the relevance of the findings for screening and breeding practices, which is important for practical applications in veterinary genetics.

However, there are some areas for improvement that should be noticed. First of all, while the abstract mentions previous findings from US and European cohorts, it could benefit from a brief explanation of why these specific mutations were chosen and their relevance to DCM. Secondly, the sample size of 74 dogs may be considered small for genetic studies. A statement addressing the potential limitations of this sample size and its impact on the generalizability of the results would strengthen the abstract. Beyond that, while p-values are provided, additional context regarding the methodology for calculating odds ratios (OR) and confidence intervals (CI) would enhance transparency and reproducibility. Eventually, it might be helpful to specify whether the mutations are novel or previously identified, as this could inform readers about the novelty of the findings.

Overall, the abstract effectively summarizes the research conducted and presents meaningful findings regarding the association of RNF207 with DCM in Dobermanns in the UK. Addressing the areas for improvement could enhance clarity and provide a more comprehensive understanding of the study's significance and limitations.

Introduction

The introduction provides a thorough overview of dilated cardiomyopathy (DCM) in dogs, particularly focusing on Dobermanns. It outlines the disease's prevalence, genetic implications, and previous research findings related to specific gene mutations. The authors set the stage for their hypothesis regarding the association of TTN, PDK4, and RNF207 with DCM in a UK cohort.

The introduction effectively establishes the significance of DCM in Dobermanns, including its prevalence and impact on health, which helps readers understand the importance of the study.

The authors provide a well-rounded review of existing literature, discussing both US and European studies that have explored genetic associations with DCM. This contextualizes their research within the broader scientific conversation.

However, here are some specific comments and suggestions to enhance the introduction further:

Several studies, including the RNF207 gene's potential role in DCM, are mentioned but lack detailed explanations. The transition between previous findings and the hypothesis could be smoother. It is suggested to include specific research objectives or questions to guide the study beyond gene sequencing for a clearer narrative flow.

Materials and Methods

Here are some comments and suggestions regarding the "Materials and Methods" section of your study:

The multicenter, retrospective case-control design is clearly stated, which helps readers understand the framework of the study.

Mentioning the ethical review committee and approval number adds credibility and shows adherence to ethical standards.

The detailed diagnostic criteria for DCM, including specific measurements and thresholds, is well-articulated and demonstrates rigor in identifying cases.

The DNA extraction process is adequately described, including the use of specific kits and methods for quality assessment.

There is no mention of statistical methods used for analysis. This is crucial for understanding how results will be interpreted.

Results

With regard to positive aspects: the study provides a detailed description of the study population, including the selection criteria and demographic information (e.g., age, sex distribution). This enhances transparency and reproducibility.

The results present specific echocardiographic measurements that highlight significant differences between DCM and control groups. This is crucial for understanding the physiological implications of DCM in Dobermanns.

The inclusion of allele frequencies for the TTN, PDK4, and RNF207 variants adds valuable genetic context to the study, which is important for understanding potential genetic predispositions to DCM.

When it comes to negative points: the study's sample size (74 Dobermanns) may limit the generalizability of the findings. Small sample sizes can lead to type II errors, where true associations are missed.

Lack of Longitudinal Data: The cross-sectional nature of the study does not allow for insights into disease progression or the effects of potential interventions over time.

Limited Genetic Analysis: While allele frequencies are reported, the association analyses do not seem to demonstrate a significant link between genetic variants and DCM, which may warrant further investigation or different analytical approaches.

Practical Comments:

Consider Increasing Sample Size: Future studies could benefit from a larger sample size to enhance the power of statistical analyses and improve the reliability of findings regarding genetic associations with DCM.

Longitudinal Studies: Conducting longitudinal studies could provide insights into how DCM progresses over time in Dobermanns and how genetic factors might influence this progression.

Broaden Genetic Analysis: Expanding the analysis to include additional genetic variants or employing whole-genome sequencing could uncover more significant associations with DCM.

Discussion

The study commences with a well-defined objective, which effectively sets the stage for the ensuing findings and provides a solid context for the results. The authors make pertinent comparisons between human and canine dilated cardiomyopathy (DCM), underscoring the differences in genetic links and their implications for understanding the disease across both species. This enriches the discussion and highlights the importance of canine models in human research.

The authors successfully place their findings within the wider framework of genetic research on DCM, referencing prior studies and established genetic mutations related to the disease in both dogs and humans.

However, there are some points that should be addressed. Although the association with RNF207 is acknowledged, there is insufficient exploration of its biological significance or the possible mechanisms through which it may impact DCM in Dobermanns. A more in-depth discussion regarding how this variant could influence cardiac function would enhance this section. In addition, the discussion fails to provide explicit recommendations for future research or how the findings could guide breeding practices, genetic screening, or clinical interventions. This omission may leave readers seeking more definitive next steps.

Reviewer #2: This well-written article makes a valuable contribution to animal cardiogenetics, offering insights relevant to both dogs and humans with dilated cardiomyopathy. Below are a few points that may enhance the manuscript's clarity and impact.

1. Have the authors considered whether regional breeding practices might influence the prevalence of certain DCM-associated mutations within the UK cohort compared to other European or U.S. populations? This could be expanded by discussing the possibility of founder effects, especially given the historically limited genetic diversity within the Dobermann breed.

2. Could the authors clarify if there were any specific subgroups or characteristics within the DCM cohort that influenced the strength of the RNF207 association? Were there phenotype variations based on the homozygous presence of the RNF207 variant?

3. Did the authors perform a power analysis to determine if this sample size was adequate to detect statistically significant differences, especially given the allele frequencies presented? It would strengthen the study to discuss the potential limitations related to sample size, especially for low-frequency variants.

4. Are chi-square tests appropriate given the sample size, particularly for rare genotype combinations? Did the authors consider alternative statistical methods (e.g., Fisher’s exact test) for smaller samples?

5. While the abstract suggests some genetic differences across populations, it would be valuable if limitations regarding population sampling, genetic heterogeneity, and broader applicability were noted. Could the authors acknowledge any potential biases in their cohort selection or limitations regarding sample representativeness?

6. Could the authors draw further parallels between the mechanisms of action of these mutations in humans and dogs, especially concerning how TTN and RNF207 may differentially influence DCM phenotypes?

6. PLOS authors have the option to publish the peer review history of their article (what does this mean? ). If published, this will include your full peer review and any attached files.

**Do you want your identity to be public for this peer review?** For information about this choice, including consent withdrawal, please see our Privacy Policy .

Reviewer #1: No

Reviewer #2: **Yes: ** Saeideh Kavousi

---

## [Author Response · Author response to Decision Letter 0]

21 Jan 2025

Reviewer #1: Abstract

The abstract presents a study investigating the association of specific genetic mutations with dilated cardiomyopathy (DCM) in Dobermanns, focusing on three genes: TTN, PDK4, and RNF207. The authors hypothesized that the TTN and PDK4 mutations would not be associated with DCM in a UK cohort, while expecting an association for the RNF207 mutation. The study involved 74 dogs, including both controls and DCM-affected animals.

Undoubtedly, the hypothesis is clearly stated, providing a solid foundation for the study and the methods are appropriate for the research question. In addition, the results are presented in a concise manner, highlighting the key findings regarding allele frequencies and statistical significance and finally in conclusion you emphasize the relevance of the findings for screening and breeding practices, which is important for practical applications in veterinary genetics.

Thank you for your comments on our study, we are pleased to hear that you found the manuscript clearly written and of importance to veterinary genetics. We also thank you for your criticism on certain aspects and have addressed these in the document below. We believe the manuscript is now stronger because of your review and comments and would of course be happy to address any of the points further if additional clarity is required.

However, there are some areas for improvement that should be noticed. First of all, while the abstract mentions previous findings from US and European cohorts, it could benefit from a brief explanation of why these specific mutations were chosen and their relevance to DCM.

Thank you for the comment regarding the explanation of the specific mutations and why they were chosen, we have included a sentence to include this in brief detail (lines 19-21), however further elaboration is limited by the word count. The explanations are otherwise included in the introduction and discussion (introduction paragraphs 2 and 3, discussion paragraphs 2, 3 and 4).

Secondly, the sample size of 74 dogs may be considered small for genetic studies. A statement addressing the potential limitations of this sample size and its impact on the generalizability of the results would strengthen the abstract

Many thanks for this suggestion and we agree with this limitation, we have added to the abstract to highlight this as suggested (line 37).

Beyond that, while p-values are provided, additional context regarding the methodology for calculating odds ratios (OR) and confidence intervals (CI) would enhance transparency and reproducibility.

We agree this would add additional reproducibility however whilst we have included OR and CI in the abstract, the methodology for these are presented in the Materials and Methods rather than the abstract as the abstract is already over word-count with the other additional suggestions.

Eventually, it might be helpful to specify whether the mutations are novel or previously identified, as this could inform readers about the novelty of the findings

We have clarified that the TTN and PDK4 variants were previously identified (line 35).

Overall, the abstract effectively summarizes the research conducted and presents meaningful findings regarding the association of RNF207 with DCM in Dobermanns in the UK. Addressing the areas for improvement could enhance clarity and provide a more comprehensive understanding of the study's significance and limitations.

Introduction

The introduction provides a thorough overview of dilated cardiomyopathy (DCM) in dogs, particularly focusing on Dobermanns. It outlines the disease's prevalence, genetic implications, and previous research findings related to specific gene mutations. The authors set the stage for their hypothesis regarding the association of TTN, PDK4, and RNF207 with DCM in a UK cohort.

The introduction effectively establishes the significance of DCM in Dobermanns, including its prevalence and impact on health, which helps readers understand the importance of the study.

The authors provide a well-rounded review of existing literature, discussing both US and European studies that have explored genetic associations with DCM. This contextualizes their research within the broader scientific conversation.

However, here are some specific comments and suggestions to enhance the introduction further:

Several studies, including the RNF207 gene's potential role in DCM, are mentioned but lack detailed explanations

We have added further explanation over the potential role of RNF207 in DCM to improve clarity and context for the reader as suggested. The lines 63-68 now read ‘RNF207 encodes a heart-specific really interesting new gene (RING) finger protein that has been shown to interact with the voltage-dependent anion channel (VDAC) 1, regulate the repolarizing channel HERG and is involved with pathological cardiac hypertrophy [14–16]. Therefore, variants in RNF207 may result in QT prolongation, shortened action potential duration and altered energy metabolism, which may play a role in DCM.’

The transition between previous findings and the hypothesis could be smoother. It is suggested to include specific research objectives or questions to guide the study beyond gene sequencing for a clearer narrative flow.

Thank you for the suggestion, we have added the following lines to the paragraph, which hopefully makes the transition smoother: ‘The previously reported TTN and PDK4 have failed to show an association with DCM in European Dobermanns and the newly discovered RNF207 mutation has not been studied in a UK cohort of Dobermanns.’ (lines 70-72).

Materials and Methods

Here are some comments and suggestions regarding the "Materials and Methods" section of your study:

The multicenter, retrospective case-control design is clearly stated, which helps readers understand the framework of the study.

Mentioning the ethical review committee and approval number adds credibility and shows adherence to ethical standards.

The detailed diagnostic criteria for DCM, including specific measurements and thresholds, is well-articulated and demonstrates rigor in identifying cases.

The DNA extraction process is adequately described, including the use of specific kits and methods for quality assessment.

There is no mention of statistical methods used for analysis. This is crucial for understanding how results will be interpreted.

Thank you for the positive review of M&Ms section and we are pleased to hear that our methods were clear. There is a section on statistical methods used (lines 154 – 162), please let us know if this requires further clarity.

Results

With regard to positive aspects: the study provides a detailed description of the study population, including the selection criteria and demographic information (e.g., age, sex distribution). This enhances transparency and reproducibility.

The results present specific echocardiographic measurements that highlight significant differences between DCM and control groups. This is crucial for understanding the physiological implications of DCM in Dobermanns.

The inclusion of allele frequencies for the TTN, PDK4, and RNF207 variants adds valuable genetic context to the study, which is important for understanding potential genetic predispositions to DCM.

When it comes to negative points: the study's sample size (74 Dobermanns) may limit the generalizability of the findings. Small sample sizes can lead to type II errors, where true associations are missed.

We agree with this point, and unfortunately due to the retrospective nature of the study there were many samples in the DNA archive that we could not rely upon to be robust for including in the study, therefore we opted to include only cases over a certain age with a confirmed diagnosis to obtain the cleanest case and control population. We have added these limitations to our discussion so it now reads: ‘The sample size was small for a genetics study, which may limit the generalizability of the results, and lead to type II errors where true associations are missed, which are important factors when interpreting our findings. Studies that are prospective using a larger population of Dobermanns would therefore be needed to confirm these findings.’ (lines 361-365).

Lack of Longitudinal Data: The cross-sectional nature of the study does not allow for insights into disease progression or the effects of potential interventions over time.

We agree that longitudinal data would be powerful to assess disease progression and effects of any interventions, however the aim of the study was to initially assess if there was credibility in the genetic variants that have been previously reported. In our limitations we mention the age cut-off of 7 years may mean some control dogs would go on to develop DCM. A very useful study to perform would be to genotype young Dobermanns and follow them over time, however given the limitation of veterinary medicine with regards to cost, owner compliance etc. this study has not been performed, but with more studies such as ours indicating which mutations need to be assessed further, hopefully future studies will provide this data.

Limited Genetic Analysis: While allele frequencies are reported, the association analyses do not seem to demonstrate a significant link between genetic variants and DCM, which may warrant further investigation or different analytical approaches.

We agree that our limited genetic analyses do not highlight strong links between variants and DCM, however there have been previously very well powered studies that used advanced genetic association techniques to find the variants we have analysed in our study, therefore it is unlikely that with our smaller sample size, additional analyses would have identified any new genetic variants. (Niskanen JE, Ohlsson Å, Ljungvall I, Drögemüller M, Ernst RF, Dooijes D, et al. Identification of novel genetic risk factors of dilated cardiomyopathy: from canine to human. Genome Med. 2023;15: 73. doi:10.1186/s13073-023-01221-3).

Practical Comments:

Consider Increasing Sample Size: Future studies could benefit from a larger sample size to enhance the power of statistical analyses and improve the reliability of findings regarding genetic associations with DCM.

Thank you for the comment and we agree that a greater sample size would be beneficial. However, in veterinary studies our sample size is similar or greater than those in the previously reported US studies. For example, Meurs et al. 2012 that identified the PDK4 mutation published in Human Genetics used 66 Dobermanns, and the 2020 paper looking at TTN and PDK4 together analysed 48 Dobermanns. We feel that despite the limitations mentioned, our study shows that the RNF207 variant appears associated with DCM in a UK population of Dobermanns which was the main objective of the study. We hope this result will support future prospective studies involving a greater sample size.

Longitudinal Studies: Conducting longitudinal studies could provide insights into how DCM progresses over time in Dobermanns and how genetic factors might influence this progression.

We agree that longitudinal studies following those genetically at risk would be beneficial, and one we are considering. However due to the numbers required this would need a multicentre approach to recruit the number of cases and significant funding (due to the age-related penetrance of the disease and the need to wait 10-15 years until all the dogs had died). Our aim was first to analyse retrospective data we had available to first ascertain which mutations are likely to be associated and would benefit from longitudinal studies.

Broaden Genetic Analysis: Expanding the analysis to include additional genetic variants or employing whole-genome sequencing could uncover more significant associations with DCM.

We agree this would be a powerful approach, but this was not the aim of the study, especially considering the small sample size. Our study was specifically looking at variants that had already been identified using the techniques you describe.

Discussion

The study commences with a well-defined objective, which effectively sets the stage for the ensuing findings and provides a solid context for the results. The authors make pertinent comparisons between human and canine dilated cardiomyopathy (DCM), underscoring the differences in genetic links and their implications for understanding the disease across both species. This enriches the discussion and highlights the importance of canine models in human research.

The authors successfully place their findings within the wider framework of genetic research on DCM, referencing prior studies and established genetic mutations related to the disease in both dogs and humans.

However, there are some points that should be addressed. Although the association with RNF207 is acknowledged, there is insufficient exploration of its biological significance or the possible mechanisms through which it may impact DCM in Dobermanns. A more in-depth discussion regarding how this variant could influence cardiac function would enhance this section.

Thank you for this comment, although the exact role of RNF207 in cardiac function and DCM is still being explored, we have added additional detail to this section so it now reads (lines 320 – 343): ‘RNF207 encodes the really interesting new gene (RING)-finger protein 207 (RNF207), which has proposed actions in regulating the cardiac action potential and is involved in pathways known to regulate cardiac hypertrophy [15,16]. Additionally, it has been shown in humans that abnormal function of the RNF207 protein is implicated in or can worsen the expression of long-QT syndrome, known to result in ventricular arrhythmias causing syncope and sudden death [36,37]. Also it has been shown that RNF207 plays an important role in the production of ATP in cardiomyocytes, and that dysfunction in this protein may worsen the development of congestive heart failure [14]. In a study using canine myocardium, researchers discovered that RNF207 is expressed in the intercalated discs between cardiomyocytes (as well as in the cytoplasm and perinuclearly) [11] . Since the gap junctions present in the intercalated discs are vital for ion passage between cells and coordinated cardiac contraction, disruption to this region may lead to impaired cardiac function. Other studies have shown that alteration to the intercalated discs is directly associated with the development of DCM [38–40] . Additionally, immunohistochemistry in a dog that was homozygous for the RNF207 variant showed marked cellular mosaicism for RNF207, indicating the variant alters expression of the protein that may affect cardiac function by the previously mentioned mechanisms [11]. Additionally, due to the role of RNF207 in ATP production in cardiomyocytes, it is also possible that mutations in RNF207 could influence the expression of DCM in Dobermanns as opposed to being directly causal, and this would also explain why not all Dobermanns with DCM carry this variant.’

In addition, the discussion fails to provide explicit recommendations for future research or how the findings could guide breeding practices, genetic screening, or clinical interventions. This omission may leave readers seeking more definitive next steps

We have updated our recommendation on further research to me more specific so this reads ‘Further research is warranted that validates the findings for the variant in RNF207 using larger sample sizes with prospective study design and following dogs longitudinally to assess outcome in patients with certain genotypes. Additional research could then assess whether early detection of at-risk individuals and therefore earlier clinical intervention (such as more regular screening or treatment with disease-modifying drugs) may improve outcomes in this patient population.’ (lines 383 - 389).

Reviewer #2: This well-written article makes a valuable contribution to animal cardiogenetics, offering insights relevant to both dogs and humans with dilated cardiomyopathy. Below are a few points that may enhance the manuscript's clarity and impact.

Thank you for your comments on our study, we are pleased to hea

---

## [Decision Letter · Decision Letter 1]

11 Feb 2025

Association of the TTN, PDK4, and RNF207 mutations with dilated cardiomyopathy in Dobermanns from the United Kingdom

PONE-D-24-45194R1

Dear Dr. Dutton,

We’re pleased to inform you that your manuscript has been judged scientifically suitable for publication and will be formally accepted for publication once it meets all outstanding technical requirements.

Kind regards,

Nejat Mahdieh

Academic Editor

PLOS ONE

Additional Editor Comments (optional):

Reviewers' comments:

Reviewer's Responses to Questions

**Comments to the Author**

1. If the authors have adequately addressed your comments raised in a previous round of review and you feel that this manuscript is now acceptable for publication, you may indicate that here to bypass the “Comments to the Author” section, enter your conflict of interest statement in the “Confidential to Editor” section, and submit your "Accept" recommendation.

Reviewer #3: All comments have been addressed

2. Is the manuscript technically sound, and do the data support the conclusions?

Reviewer #3: Yes

3. Has the statistical analysis been performed appropriately and rigorously? 

Reviewer #3: Yes

4. Have the authors made all data underlying the findings in their manuscript fully available?

Reviewer #3: Yes

5. Is the manuscript presented in an intelligible fashion and written in standard English?

Reviewer #3: Yes

6. Review Comments to the Author

Reviewer #3: Thank you for submitting your manuscript for review. I appreciate the effort and time you dedicated to this work. I have carefully evaluated the paper and noted that you have made a commendable effort to address all the comments provided.

It is evident that you have taken the feedback into account, and I acknowledge the revisions made to improve the clarity and quality of the manuscript.

Should you have any further questions or require additional feedback, please feel free to reach out.

7. PLOS authors have the option to publish the peer review history of their article (what does this mean? ). If published, this will include your full peer review and any attached files.

**Do you want your identity to be public for this peer review?** For information about this choice, including consent withdrawal, please see our Privacy Policy .

Reviewer #3: No

---

## [Editor Report · Acceptance letter]

PONE-D-24-45194R1

PLOS ONE

Dear Dr. Dutton,

I'm pleased to inform you that your manuscript has been deemed suitable for publication in PLOS ONE. Congratulations! Your manuscript is now being handed over to our production team.

Kind regards,

on behalf of

Dr. Nejat Mahdieh

Academic Editor

PLOS ONE